# Short-Term Fish Oil Supplementation during Adolescence Supports Sex-Specific Impact on Adulthood Visuospatial Memory and Cognitive Flexibility

**DOI:** 10.3390/nu14173513

**Published:** 2022-08-26

**Authors:** Julie Raymond, Alexandre Morin, Marilou Poitras, Hélène Plamondon

**Affiliations:** Behavioural Neuroscience Group, School of Psychology, University of Ottawa, 136 Jean-Jacques Lussier, Ottawa, ON K1N 6N5, Canada

**Keywords:** *n*-3 PUFA, docosahexaenoic acid, eicosapentaenoic acid, sex-dependent, age-dependent, Barnes Maze Test, visuospatial memory, cognitive flexibility

## Abstract

Numerous studies have supported benefits of omega-3 supplementation using Menhaden fish oil (FO) to promote brain maturation and plasticity during critical developmental periods. The goal of this study was to determine sex-specific immediate and delayed impact of adolescent omega-3 supplementation on visuospatial memory and cognitive flexibility. Sixty-four Wistar rats (*n* = 32 males and females) received daily FO or soybean oil (CSO) supplementation via oral gavage (0.3 mL/100 g body weight) from postnatal day 28–47. The Barnes Maze Test (BMT) was used to measure visuospatial memory and reversal learning trials (RL) determined cognitive flexibility. Juveniles underwent testing immediately after the gavage period, while adults began testing on postnatal day 90. Adult rats showed reduced working memory errors (WME) and gradual decrease in escape latencies compared to juveniles. Importantly, adult FO-supplemented females displayed fewer WME than males, while males’ performance benefited from CSO supplementation. Overall, sex- and supplementation-dependent effects supported a positive impact of FO in female rats only. Our findings support the potential for supplementation limited to the early adolescence period to influence adulthood spatial learning and cognitive flexibility in a sex-specific manner.

## 1. Introduction

Dietary habits have significantly evolved over the last century, mainly due to industrialisation which revolutionised agricultural practices [1], and a switch from consumption of untransformed foods such as nuts, fish, and vegetables to processed grain-based diets with higher fat and reduced fibre content. Expansion of the so-called Western diet led to a substantial increase in omega-6 polyunsaturated fatty acid (*n*-6 PUFAs), saturated fat, and trans-fat dietary intake at the expense of essential fatty acids (EFA) such as omega-3 polyunsaturated acids (*n*-3 PUFAs) [2]. *n*-3 PUFAs are mainly found in plants and nuts, with highest levels being metabolised from oily substances like sunflower oil, salmon, herring, and mackerel [3,4]. The disproportionate consumption of *n*-6 fatty acids in the Western diet compromises the *n*-6 to *n*-3 ratio, which can then reach a 20:1 ratio [3,5], significantly exceeding the optimal 4:1 ratio [1]. On the other hand, the nut- and fish-rich Mediterranean diet, in which *n*-3 is highly present [6], is known for having numerous health benefits [7,8], showcasing the importance of *n*-3 PUFAs.

A major concern about societal dietary shifts comes from the fact that *n*-3 PUFAs cannot be synthesised by the human body and therefore must be acquired through diet [9]. In this context, dietary deficits have generated negative impacts on brain maturation processes and associated functional outcomes [9]. Indeed, raised endogenous levels of docosahexaenoic (DHA) and eicosapentaenoic (EPA) acids brought by omega-3 consumption are related to a variety of essential roles in humans and animals, notably promotion of membrane fluidity and synapse formation in the brain, favoring vesicle formation and protein transport [10,11], and improvement of cognitive and attentional capacities [9,12,13,14]. Discrete studies have supported DHA supplementation to improve cognitive development in children [15,16,17] and provide beneficial effects on cognitive task performance in juvenile rats [18,19,20,21]. Evaluating the impact of daily DHA enriched diet for underperforming children, Richardson et al. [16] showed supplementation between the age of 7 and 9 to improve overall reading performance. Likewise, Dalton et al. [22] highlighted *n*-3 PUFA dietary intake to enhance verbal learning and memory in children of the same ages. Studies also reported maternal omega-3 intake to influence toddlers IQ scores [23,24]. Notably, a study by Lassek & Gaulin [24] provided compelling sex-dependent effects of an *n*-3 PUFA enriched diet, reporting measurable impact of *n*-3 PUFA intake on cognitive test scores in both sexes, although greater advantage was observed in females. Similar observations were made in rodents. Consistently, Fedorova et al. [19,25] have shown that maternal PUFAs deficiency during the gestational and lactational periods (7–8 weeks) led to impaired performance on the Barnes Maze test (BMT) in Long Evans rats and in mice. These rats also demonstrated reduced reversal learning (RL), validating the influence of omega-3 on cognitive flexibility. Later life cognitive abilities are also influenced by the consumption of omega-3, with DHA consumption protecting the hippocampus from degeneration and delaying cognitive decline in both humans and non-human mammals [26].

Although several studies have supported the influence of in utero, early developmental, and adult omega-3 dietary intake on cognition, the impact of adolescent *n*-3 PUFA supplementation on cognitive processes remains to be established [23,27,28]. This developmental stage represents a critical time window marked by extensive brain maturation, affecting the “late blooming” prefrontal cortex (PFC) for which the course of maturation is prolonged well after puberty [29,30,31,32,33]. The crucial role of neural outputs from the frontal cortex to areas such as the hippocampus, basal ganglia, cerebellum, thalamus, and other association cortices in cognitive processes, executive function, and reversal learning is well acknowledged [34,35]. Importantly, the PFC is the region which stores the highest DHA concentrations. Indeed, of the 35–40% DHA concentrations stored in the brain, more than 15% accumulates in the frontal cortex [21,36].

Several studies have explored the association between *n*-3 PUFA supplementation and improved cognition. For instance, *n*-3 PUFA has been shown to improve hippocampal-dependent learning, adulthood neurogenesis, and facilitate learning processes [37,38]. Not only has DHA supplementation been shown to increase cognitive performance, but researchers also report its importance in the attention process itself. Yoshida et al. [37] demonstrated that reduced α-Linolenate consumption, a precursor of DHA, led to a nearly 30% decrease of synaptic vesicle density in the terminals of the hippocampal CA1 region and created learning impairments (e.g., increased error rate) in the brightness-discrimination learning test. On a similar note, Ahmad et al. [39] demonstrated a decrease of cell density, volume and total cell counts in the CA1 and CA3 regions of the hippocampus associated with a high dietary intake of linoleic acid (LA), likely to contribute to observed spatial memory deficits.

Interestingly, effective *n*-3 PUFA conversion is influenced by several factors, notably gonadal hormones. Testosterone has been found to inhibit adequate conversion of *n*-3 PUFAs while estrogen promotes the separation of free fatty acids toward oxidation and away from triglyceride storage [40,41,42]. Elevated levels of testosterone during development have been proposed to influence negatively attentional abilities and sociability in children [43], whereas estrogen is known to exert beneficial effects on cognition, although the relationship with *n*-3 PUFA metabolism has not been established [42,44,45]. Sexually dimorphic responses to omega-3 supplementation have been previously demonstrated, which could be of particular importance when supplementation occurs during the adolescence period [43].

### Goal and Hypothesis

Thus, the goals of this study are to: (1) characterise visuospatial memory performance in juvenile and adult rodents; (2) monitor sex-dependent effects of diet on such cognitive abilities; and (3) determine the short and long-term effects of omega-3 on visuospatial memory and cognitive flexibility. As literature identifies sex-dependent effects of omega-3 on cognitive and visuospatial performance, we expect females to show increased spatial memory performance [i.e., find the escape box faster with fewer errors], both in the juvenile and adult testing periods.

## 2. Methods

### 2.1. Subjects

Wistar rats (*n* = 64; 32 females and 32 males) arrived at the facility on postnatal day (PND) 23 and were acclimated for a period of 5 days prior to initiation of gavage procedures. Animals were supplemented daily with a high source of omega-3 through Menhaden fish oil or a control soybean oil from PND28 to PND47, covering the early and mid-adolescence period (see Figure 1 for experimental timeline). Half of each supplementation group underwent behavioural testing immediately following diet (*n* = 8 females; *n* = 8 males per condition), while the other half groups were kept in the animal facility, on a regular diet, until behavioural testing during adulthood (PND90; *n* = 8 females; *n* = 8 males per condition). Rats were kept on a 12 h light/dark cycle (lights on at 7 a.m.) and had free access to regular food chow and water throughout the experiment. Vaginal cytology was performed on adult females from PND83 to 96 to determine stages of the estrous cycle. Microscopic analysis was performed for the assessment of each stage, using guidelines by McLean et al. [46] and Goldman et al. [47]. All procedures were carried out in accordance with the Canadian Council of Animal care (CCAC) and approved by the University of Ottawa Animal Care Committee. Experimentation complied with the ARRIVE guidelines and with the National Institutes of Health guide for the care and use of laboratory animals (NIH Publications No. 8023, revised 1978).

### 2.2. Supplementation and Weight Monitoring

Menhaden fish oil (FO; Sigma-Aldrich, Oakville, ON, Canada) and soybean oil (CSO; Sigma-Aldrich, Oakville, ON, Canada) were administered via oral gavage at a concentration of 0.3 mL of supplement per 100 g of body weight [48]. Supplementations were composed of the following: (1) Control-Soybean oil (CSO), a non-hydrogenated omega-6-rich supplement with a 7.4/1 *n*-6/*n*-3 ratio and (2) Fish oil from menhaden fish (FO), an omega-3-rich supplement (Sigma Aldrich; 20–31% omega-3 fatty acids content; EPA—10–15%, DHA 8–15%), and administered daily between 7 and 9 a.m. To facilitate the procedure and limit gavage-induced stress, the tip of gavage syringes was slightly dipped in condensed milk (Eagle Brand^®^ Low Fat Sweetened Condensed Milk). Weight was monitored daily during oral gavage, and once per week until euthanasia to ensure proper weight gain and development.

### 2.3. Behavioural Testing

On testing days, rats were moved from the vivarium to an acclimation room near the testing area at 7 a.m. for a minimum of 30 min prior to testing. Females’ estrous cycle was assessed at least 1 h before behavioural testing. The testing room was kept between 21–23 °C with 40–60% humidity and brightly lit with 400 lux overhead lights. A black curtain was used to separate the testing area from the researcher. The testing apparatus was cleaned with 70% ethanol between each animal. Every test was recorded using a ceiling-mounted camera (Panasonic^®^ Analog Camera, Model: WV-CP284) and coded using ODlog^TM^ (V2.x, Macropod Software, Abbotsford, Australia) and Noldus Ethovision video tracking (Noldus, Leesburg, VA, USA).

#### Cognitive Testing: Barnes Maze Test (BMT) and Reversal Learning (-RL)

The BMT was used to assess spatial memory in juvenile and adult rats [49]. The apparatus consisted of a plastic circular slab (d = 122 cm; c = 376.8 cm) with 18 equidistant holes (d = 10 cm). An escape box was attached under one hole using a drawer-like system. Geometrical shapes placed on a black curtain surrounding the maze acted as visuospatial cues. Two bright lights (400 lux) illuminated the arena, increasing the rats’ motivation to find the escape box. Trials were conducted over the course of 6 to 9 days [three additional days were required when reversal learning was assessed], with 2 daily trials lasting a maximum of 5 min each. Rats were placed under an opaque plastic container in the middle of the maze and the container was lifted via a pulley system to start the exploration period. Each time the rat found the escape box, it was kept in the box for a period of 90 s before being placed back in its cage. The second trial took place on average 3 h following the first trial. If the rat did not enter the escape box within 5 min, the experimenter gently guided the animal to the escape box and encouraged entry. The location of the escape box remained the same throughout the 6 days of acquisition. In adult groups, three additional daily sessions evaluated reversal learning. In those sessions, the testing procedure was identical with the exception that the escape box was positioned under a hole which differed from the one used during acquisition.

Latency to find the escape box (s) and the number of working memory errors (WME; poking their nose in hole not associated with the escape box) were measured using ODlog^TM^ software (V2.x, Macropod Software, Abbotsford, Australia). Distance travelled (cm) before entry in the escape box was recorded and analysed using Noldus EthoVision video tracking system (Noldus, Leesburg, VA, USA).

### 2.4. Statistical Analyses

For the acquisition phase, five-way mixed analyses of variance (ANOVA) were performed using IBM SPSS Statistics 28.0 software (IBM, Armonk, NY, United States) with between-group factors Supplementation (FO vs. CSO), Age (juvenile vs. adult), and Sex (female vs. male) as well as within-group factors Day (1–6) and Trial (1 vs. 2). Four-way mixed ANOVA was performed for reversal learning trials, with Supplementation (FO vs. CSO) and Sex (females vs. males) as between-group factors and Day (1 to 3) and Trial (1 vs. 2) as within-group factors. The outliers were corrected by using the group’s now most extreme value plus or minus 1, for upper- and lower-end data, respectively. All assumptions were met prior to analyses. In case of violation of sphericity, Greenhouse-Geisser correction was used. Effect size was computed alongside alpha value (α ≤ 0.05). For all measures, significance was set at *p* < 0.05 and Bonferroni correction was applied for all pairwise comparisons. 

## 3. Results

### 3.1. Acquisition Phase: Latency to Escape Box Entry

For latency to reach the escape box, five-way mixed ANOVA revealed main effects of testing Days and Trials as well as Day*Age and Trial*Sex interactions (See Table 1 for ANOVA summary). No impact of dietary supplementation was observed. Pairwise comparisons for Day*Age analysis revealed juvenile rats to require increased latencies to find the escape box compared to adult rats, apparent on DAY4 (*p* = 0.044), 5 (*p* = 0.014), and 6 (*p* = 0.003; see Figure 2A). Furthermore, juvenile rats showed reduced latencies on DAY1 compared to DAY4 (*p* = 0.038), 5 (*p* = 0.013), and 6 (*p* = 0.003), which contrasted the gradually reducing latencies observed over days in adult rats, and evidenced between DAY1 and 3 (*p* = 0.002), 4, 5, and 6 (all *p* < 0.0005), DAY2 and 4 (*p* = 0.01), 5, and 6 (both *p* < 0.0005), DAY3 and 5 (*p* = 0.004) and 6 (*p* < 0.0005) as well as between DAY4 and 5 (*p* = 0.039) and 6 (*p* < 0.0005). Post hoc comparisons examining the Trial*Sex interaction highlight both female and male rats to show increased latency during Trial 1 than during Trial 2 (*p* = 0.004 for females; *p* < 0.0005 for males).

### 3.2. Acquisition Phase: Distance Travelled

Five-way mixed ANOVA revealed significant main effects of Day and Trial (see Table 1). Significant interactions were found for Day×Sex, Day×Age, Trial×Supplementation, Day×Trial, and Trial×Supplementation×Sex×Age (See Figure 3). Two rats were omitted from this analysis due to technical issues and their data points were replaced using multiple imputation.

Pairwise comparisons for the Day*Sex interaction supported increased distance travelled by females compared to males on DAY1, 2 (*p* < 0.0005 for each comparison), and 4 (marginal *p* = 0.05). However, both sexes showed reduced distance travelled as testing days progressed (*p* < 0.05; see Figure 3A). The Day*Age interactions was attributable to longer distance travelled in juveniles compared to adults on DAY4 (*p* < 0.0005), 5 (*p* = 0.007), and 6 (*p* = 0.001). Juvenile rats also travelled more on DAY1 (*p* = 0.002), 2 (*p* = 0.003), and 4 (*p* < 0.0005) than on DAY5 and more on DAY1 through 4 than 6 (all *p* < 0.007). Adult rats similarly travelled more on DAY1 and 2 than on DAY3, 4, 5, and 6 (all *p* ≤ 0.001; see Figure 3B).

Pairwise comparisons for the Trial*Supplementation interaction supported increased distance travelled in both FO and CSO rats in Trial 1 when compared to Trial 2 (both *p* < 0.0005). As for the Day*Trial interaction, post hoc comparisons supported rats to travel longer distances during the first compared to the second daily trials from DAY2 through DAY6 (all *p* < 0.002). Similarly, travelled distance for both Trials 1 and 2 gradually decreased over testing days (all *p* < 0.035). Three-way interactions of Age, Sex and Supplementation during the first trial *F*(1,55) = 0.12, *p* = 0.728; or during the second trial *F*(1,55) = 1.30, *p* = 0.259 were not significant.

### 3.3. Acquisition Phase: Number of Working Memory Errors (WME)

For the number of WMEs, five-way mixed ANOVA revealed main effects of Day and Trial, and significant interactions for Day*Age, Day*Supplementation*Sex, Day*Sex*Age, Trial*Age, Day*Trial, Day*Trial*Age, and Day*Trial*Supplementation*Sex (See Table 1). Pairwise comparisons for Day*Age showed that adults made more WME than juveniles on DAY1 (*p* = 0.003), while thereafter juveniles made more WME compared to adults [DAY3,4, 5 and 6—all *p* < 0.0005]. The juvenile rats made fewer WME on DAY1 compared to DAY2 (*p* = 0.006), 3, 4, 5, and 6 (all *p* < 0.0005). As Figure 2B illustrates, post hoc analyses related to the Day*Supplementation*Sex interaction showed reduced WME in CSO- compared to FO-fed males (*p* = 0.017) and reduced WME in CSO males compared to female counterparts (*p* = 0.05). On DAY4, CSO males made less WME than FO males (*p* = 0.05) while FO- females made less WME than FO-fed males (*p* = 0.034).

An effect of age emerged from post hoc comparisons assessing the Trial*Age interaction, which supported reduced WME in adult compared to juvenile rats during Trial 1 and Trial 2 (both *p* < 0.005). The juvenile and adult cohorts made more WME during Trial 1 than Trial 2 (*p* = 0.044 for juveniles; *p* < 0.0005 for adults). For the Day*Trial interaction, pairwise comparisons showed more WME were committed during Trial 1 on DAY5 than on DAY6 (*p* = 0.011). There were more WME during Trial 1 than Trial 2 on DAY4 (*p* = 0.016) and DAY5 (*p* < 0.0005; see Figure 2C). Post hoc analysis of the Day*Sex*Age interaction was attributable to juvenile females making less WME than juvenile males (*p* = 0.011) on DAY4, and to adult (females and males) making less WME than juvenile counterparts (*p* = 0.002 and *p* < 0.0005, respectively; see Figure 2D). The Day*Trial*Age interaction is related to adult cohorts making less WME over days for both Trial 1 and 2 compared to juvenile cohorts (*p* < 0.025).

### 3.4. Reversal Learning: Latency to Escape Box Entry

Four-way mixed ANOVAs on adult rats’ latencies to enter the escape box during reversal learning trials revealed a main effect of Day [*F*(2,58) = 24.926; *p* ≤ 0.0005; η^2^*_p_* = 0.462], attributable to all rats showing increased entry latencies on DAY1 compared to DAY2 and 3 (both *p* < 0.0005). In addition, a trend was found for Trials [*F*(1,29) = 4.077; *p* = 0.053; η^2^*_p_* = 0.123] related to a tendency for all rats to enter the escape box quicker during the second than the first trial (*p* = 0.053). For the ANOVA summary of the reversal learning trials, please see Table 2.

### 3.5. Reversal Learning: Distance Travelled

In adults rats, four-way mixed ANOVA revealed significant main effects of Day [*F*(2,54) = 15.019; *p* < 0.0005; η^2^*_p_* = 0.357] and Trial [*F*(1,27) = 5.734; *p* = 0.024; η^2^*_p_* = 0.175] as well as significant interactions of Day*Sex [*F*(2,54) = 10.731; *p* < 0.0005; η^2^*_p_* = 0.284] and Day×Trial×Sex [*F*(2,54) = 7.113; *p* = 0.002; η^2^*_p_* = 0.209] (See Figure 4A). No effect of supplements was observed (*p >* 0.05). The Day*Sex interaction is related to males travelling longer distances than females on DAY2 (*p* = 0.011). Females travelled longer distances on DAY1 compared to DAY2 (*p* = 0.002) and 3 (*p* < 0.0005) while males’ distance travelled was longer on DAY1 (*p* = 0.003) and 2 (*p* < 0.0005) compared to DAY3. On the other hand, pairwise comparisons for Day×Trial×Sex showed females to have travelled more during the first trial on DAY1 when compared with DAY2 and 3 (*p* < 0.0005) while males exhibited such effect between DAY2 and 3 (*p* = 0.044). During Trial 2, males travelled longer distance on DAY1 than on DAY2 (*p* = 0.04) and 3 (*p* = 0.005). On DAY1 and 2, females travelled longer distances during Trial 1 compared to Trial 2 (both *p* < 0.05). In sum, travelled distance decreased over days [the longer distances being travelled on DAY1 compared to DAY2 (*p* = 0.02) and 3 (*p* < 0.0005)] and from the second compared to the first trial (*p* = 0.024).

### 3.6. Reversal Learning: Working Memory Errors

A four-way mixed ANOVA conducted in adult rats showed a significant main effect of Day [*F*(2,56) = 13.818; *p* < 0.0005; η^2^*_p_* = 0.33] and an interaction of Day*Trial*Sex [*F*(2,56) = 3.195; *p* = 0.049; η^2^*_p_* = 0.102]. No effect of supplements was observed (*p* > 0.05). For the Day*Trial*Sex interaction, post hoc analyses revealed that, during Trials 1 and 2, females made more WME on DAY1 than on DAY2 and 3 (all *p* < 0.04). Males, on the other hand, committed more WME on DAY1 than DAY3 (*p* = 0.022) during Trial 2. Overall, WME decreased over days, with rats exhibiting more on DAY1 than 2 (*p* = 0.04) and more on DAY1 and 2 than on DAY3 (both *p* < 0.02) (see Figure 4B). See Figure 5 for summary of estrous cycle stages in adult females for all testing days.

## 4. Discussion

The present study characterised sex-specific and time-limited effects of adolescent omega-3 supplementation on visuospatial memory performance and cognitive flexibility during different developmental periods. To date, researchers have reported notable impacts of omega-3 supplementation during pregnancy, adulthood or aging on cognitive function, leaving other critical developmental periods largely unassessed. Furthermore, the determination of differences in nutritional effects associated with sex, including from omega-3 supplementation, has been neglected. Our study uncovered age and sex as important factors regulating visuospatial memory formation and adulthood cognitive flexibility. Notably, our findings supported FO supplementation to be associated with increased working memory errors in males compared to females and soybean oil fed males.

### 4.1. Contrasting Visuospatial Learning Abilities in Adult and Juvenile Rodents

Our study indicated 20-day omega-3 supplementation to exert minimal influence on distance travelled or latency to escape box entry in juvenile and adult rodents. However, our findings support age-dependent differences regarding latency and distance travelled to escape box entry, with juveniles exhibiting significantly poorer performances on both measures compared to adult counterparts. One may argue that the complexity level of the BMT may exceed the abilities of juvenile rats’ hippocampal networks to process and remember such visuospatial information [50], explaining the observed age-related performances. Notably, in 2018, McHail et al. [51] developed a smaller version of the BMT aimed at assessing visuospatial learning in juveniles, where juvenile rats were exposed to a smaller testing area with fewer options (8 holes rather than 16). Their findings supported the ability of juvenile and adult rats to find the escape box and learn the spatial task, although juveniles tended to use efficient spatial search techniques from the second day compared to adult rodents being able to achieve such behaviours from the first exposure. They concluded that this simplified BMT would be appropriate for younger rats aged from PND17 to 26 [51]. In this study, rats were several weeks older than in McHail et al.’s [51] study, and according to Blair et al.’s [52] findings, juvenile rodents reach similar hippocampal maturation as adults towards the end of the third postnatal week, which falls several weeks prior to the first day of testing in our own study. Additionally, Wills et al. [53] demonstrated adult-like directional firing of grid cells from the hippocampal region linked to spatial representation, achieving maturity of this system often around PND28-29, sometimes even earlier. As such, juveniles most likely passed this critical period, and therefore, the age differences observed in our sample may not be related to an incomplete hippocampal maturation.

Contrasting an inability of juvenile rats to remember the task over days, both age groups were able to reduce escape latencies within days [i.e., from Trial 1 to Trial 2]. This observation supports the ability of juveniles to acquire the task’s rules and apply them within a short presentation delay, although not being able to maintain such learning active for time periods supporting consolidation and improvements in performance over days. In accordance with these findings, Brown & Kraemer [54] found that juvenile rodents tend to forget faster than adults and to show significant increases in latencies when testing sessions are presented irregularly. Together, these observations support juveniles’ memory consolidation pathways involving hippocampal long-term potentiation and prefrontal cortex activation to remain functionally immature, leading us to hypothesise that juveniles would show increased difficulties in the reversal learning task, which requires prefrontal activation.

### 4.2. Steady Improvements of Working Memory Performance in Adult Rats Contrasts Persistent Learning Difficulties in Juvenile Male Rats

As adults, spatial memory performance was comparable in males and females, with both sexes showing reduced WME as daily testing progressed. Interestingly, while juvenile rats as a group experienced learning difficulties, this was particularly notable in male cohorts, which showed the highest number of WME compared to all other groups (from testing days 3–5; see Figure 2D), indicating a spatial learning deficit in juvenile males. These findings contrast other studies showing male rodents to complete spatial tasks more efficiently than their female counterparts [55,56]. For instance, Jonasson [55] reported male rodents to solve spatial problems quicker and perform better overall compared to females. Many studies have linked this performance gap between males and females to differences in sex hormones [57,58]. McCarthy & Konkle [57] demonstrated that elevated testosterone secretion in male rodents around PND45, which falls a few days prior to testing in our study, can lead to improved learning and memory.

The reasons pertaining attenuated WM performance in males remain uncertain. Notably, this effect was particularly salient in FO-fed males (see Figure 2B). Although reduced performance related to *n*-3 PUFA supplementation was not expected, previous studies have observed supplementation with unsaturated fat to impair performance in male but not female guinea pigs by way of males showing increase WME and latencies in a spatial memory test (radial Y-maze; [59]). In this context, Omega-3 PUFAs are known to be differentially metabolised in male and female rodents due to testosterone reducing the conversion of PUFAs following its consumption, an effect related to reduced attention and social behaviour in boys [41,42,43]. Considering these differences, males may require more important FO concentrations than females to achieve similar benefits. Indeed, we recently demonstrated a similar FO concentration as the one in this study to elevate anxiety-like behaviour and reduced OFT and EPM exploration, while soybean oil supplementation fostered increased sociability in male compared to female rats (Raymond et al., in preparation). Similarly, Teixeira et al. [60] showed soybean oil supplements in male rodents to improve BMT performance compared to lard or hydrogenated vegetable fat supplemented groups. On a similar note, Crane & Greenwood [61] observed that a 4-week soybean oil supplementation favours cognitive performance in male mice. These findings concur with our observations and support differential benefits from FO supplementation in male and female rodents. It must be noted that studies showing positive effects of FO on memory in humans and rodents have used longer periods of supplementation, often being conducted over months [25,62,63,64]. For instance, long-term gestational DHA supplementation has been associated with increased neurite outgrowth and synaptogenesis in infants having a positive impact on learning and memory [65]. A limited period of FO supplementation, even performed during the critical adolescence maturation period, was not sufficient to significantly impact cognitive abilities in male rodents.

### 4.3. Sex-Dependent Effect on Working Memory Errors in Adults

In adult rats, our findings support sexual dimorphism in acquiring the BMT’s reversal learning task rules, wherein females reduced latencies and number of WME over the 3 testing days with no such improvements in males. Although mechanisms linked to these observations remain to be examined, research indicates that 17β-estradiol (E2) enhances working memory performance in the Radial Arm Maze [66] and spatial memory consolidation in the Morris Water Maze [67]. Studies further suggest rises in E2 levels in maturing female rats. From PND28 up to PND48-75, secretion of sex hormones gradually increases to finally reach adult levels [68]. While gonadal hormone levels were not measured, we observed that 65% of the females started the reversal learning trials in the proestrus or estrus stages (see Figure 5). As research indicates, females experience high estradiol levels at the start of the estrous cycle, a slight increase in progesterone in the proestrus and estrus phase as well as enhanced luteinizing hormones in the late proestrus [46,69,70]. These hormonal peaks are linked to improved memory retention and tend to promote functional connectivity of the hippocampus [70,71,72,73]. Therefore, it is plausible that elevated levels of circulatory E2, progesterone, and luteinizing hormone in females could have positively impacted learning and cognitive flexibility in the BMT. Furthermore, facilitating effects of E2 on spatial learning may also contribute to improved performance of juvenile females in the acquisition phase of the BMT. To confirm this, it would appear important for further studies addressing juveniles’ learning to formally link E2 concentrations in females during testing days (through monitoring or manipulation of E2 concentrations) with test performance, as well as characterise the impact of FO supplementation on E2 concentrations and bioavailability.

### 4.4. Changes in Search Strategy under Stressful Conditions in Male Rats

An interesting observation was noted in male rats on DAY5, Trial 1, specifically those fed with FO. A significant increase in WME was found in this group compared to other days and trials. This testing day was characterised by increased unforeseen noise, which could have had an acute impact on corticosterone secretion. Several studies have shown that spatial memory can be influenced by acute stress exposure [74,75,76], an effect reported in several spatial tests including the Y-Maze, Radial Arm Maze and Morris Water Maze. On the other hand, it is interesting to note that learning has been shown less affected by stress in the BMT [74], potentially creating an isolated impact of stress on WME as seen in our study. Moreover, Schwabe et al. [75] have reported a tendency for male rodents to modify their spatial strategies following an acute or chronic stressor, an observation not visible in females. In fact, following stress, males tend to rely more on intra-maze cues compared to extra-maze ones (such as the different shape signs placed on each wall), possibly leading males to make increased WME on the initial DAY5 trial.

### 4.5. Limitations

While our experimental design was carefully reviewed prior to testing, certain unforeseen variables were observed during our analysis. As previously stated, we detected a peak in WME made by FO-fed males on DAY5, trial 1. To confirm our hypothesis related to the presence of an auditory acute stressor during visuospatial memory testing, future studies could provide physiological data by measuring corticosterone levels following each BM daily sessions [74,75,76].

Additionally, we observed alterations in cognitive performance sporadically throughout testing. A positive impact of FO was observed in females on DAY2 and 4, potentially indicating that a 20-day supplementation period is too brief to induce long-term effects. In fact, most studies evaluating dietary supplements have reported changes in cognitive performance associated with longer ingestion periods (8–12 weeks are commonly used; 60–62) or evaluated changes over multiple generations [23,24,25]. Using progressively longer exposure to dietary supplementation could provide knowledge on the threshold where the supplementation confers a sustainable and consistent positive influence on spatial memory.

Lastly, our findings revealed statistical differences between males’ and females’ performances in the reversal task. Females made less WME and took reduced time to complete the reversal learning sessions compared to males. We proposed estradiol, luteinizing hormone, and progesterone to promote cognitive flexibility in females by enhancing memory consolidation and recall [70,71,72,73]. While this hypothesis is well documented in the literature [66,67,70,71,72,73], we did not measure hormonal fluctuation in this current study. Future studies should integrate these measures to confirm our current observation.

## 5. Conclusions

In summary, our findings indicated that, during acquisition of a Barnes Maze task, adult rodents make fewer WME, show reduced distance travelled and escape box entry latencies compared to their juvenile counterparts. In reversal learning trials, adult female rats show superior performance, travelling shorter distances and making less WME compared to male rats, suggesting improved cognitive flexibility. Although dietary supplementation targeting the adolescent period had minimal impact on performance, slight behavioural changes were observed, including opposite trends in the effect of FO supplementation on spatial memory in male and female rats. Thus, omega-3 supplementation seemed to be slightly more beneficial to females than males, an effect manifested through reduced WME in FO-fed females, although this supplementation length did not translate into significant effects. These findings support the possibility for nutritional status during adolescence to be one contributor to sexual dimorphism observed in behavioural responses, although further studies are necessary to clarify these effects. This can play an important role in terms of modulating responses through the synergistic actions of a multitude of environmental factors.

## Figures and Tables

**Figure 1 nutrients-14-03513-f001:**
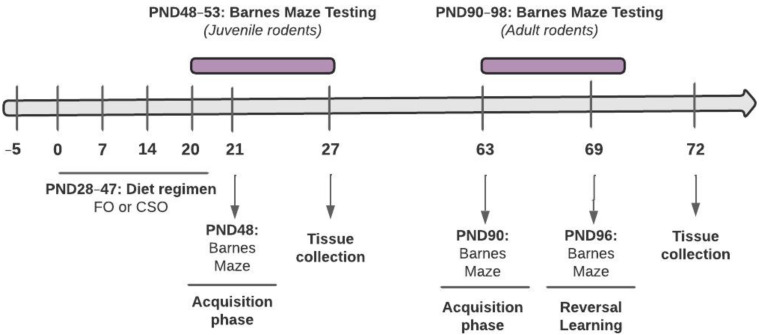
Experimental Timeline. Animals arrived at the facility on PND23 and acclimated for 5 days prior to the start of the experiment. Dietary supplementation occurred daily from PND28 to PND47 (20-day period). Half the cohort underwent Barnes Maze testing immediately following the supplementation period (*n* = 32), while the other half was tested as they reached adulthood (PND90). Barnes Maze acquisition was evaluated for 6 days in adolescent and adult rats, while reversal learning was assessed in adults only for 3 days. FO: Menhaden fish oil; CSO: Control soybean oil.

**Figure 2 nutrients-14-03513-f002:**
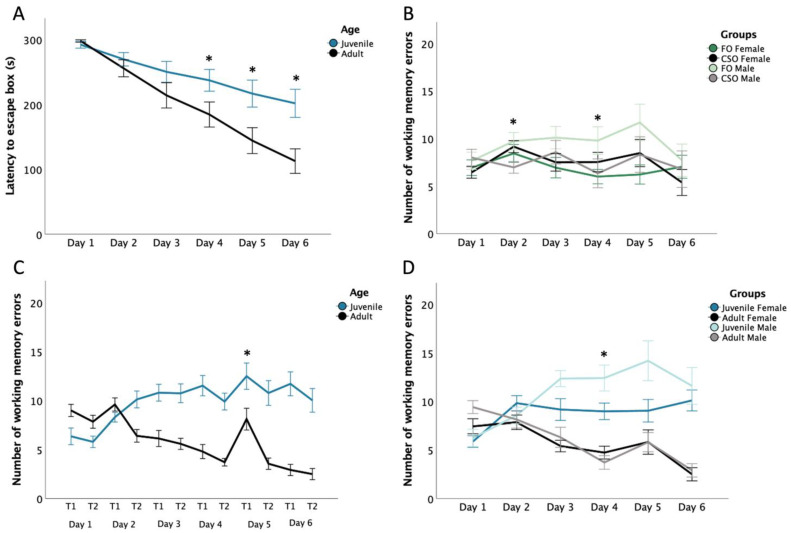
Latency to Escape Box and Number of Working Memory Errors (WME) in the Barnes Maze Test. Panel (**A**) shows the latency (s) to the escape box in juveniles and adults. Panel (**B**) shows the number of WME made by FO- and CSO-supplemented males and females. Panel (**C**) shows the number of WME made by juvenile vs. adult rats. Panel (**D**) shows the number of WME made according to rodents’ sex and age. Data are presented as mean ± S.E.M. * Indicates statistical significance at *p* < 0.05. CSO: Control soybean oil; FO: Menhaden fish oil; T1: Trial 1; T2: Trial 2.

**Figure 3 nutrients-14-03513-f003:**
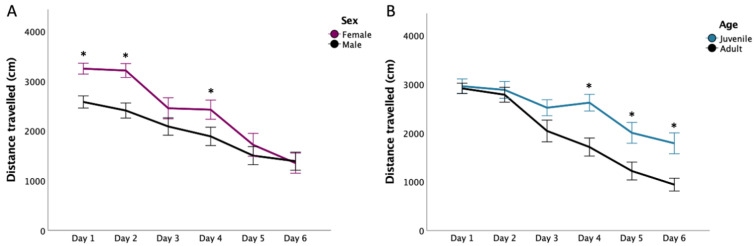
Distance Travelled to Reach the Escape Box in the Barnes Maze Test. Panel (**A**) shows the distance travelled (cm) by females and males, whereas panel (**B**) depicts the distance travelled by juveniles and adults. Data are presented as mean ± S.E.M. * Indicates statistically significant difference between groups at *p* < 0.05. T1: Trial 1; T2: Trial 2.

**Figure 4 nutrients-14-03513-f004:**
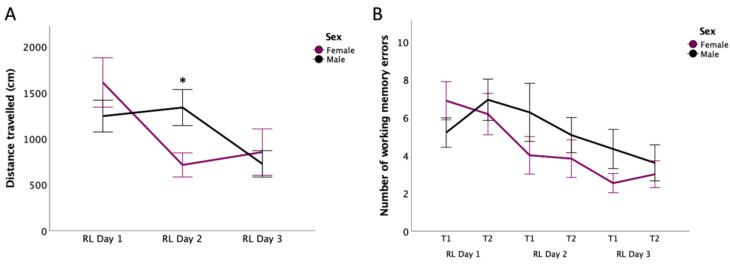
Reversal Learning Trials in the Barnes Maze Test (Adult Rats Only). Panel (**A**) shows the distance travelled (cm) for males and females during the 3 days of reversal learning. Panel (**B**) presents the number of working memory errors in male and female rats. Data are presented as mean ± S.E.M. * Indicates statistically significant difference between groups at *p* < 0.05. No effect of the supplementation was observed. RL: Reversal learning; T1: Trial 1; T2: Trial 2.

**Figure 5 nutrients-14-03513-f005:**
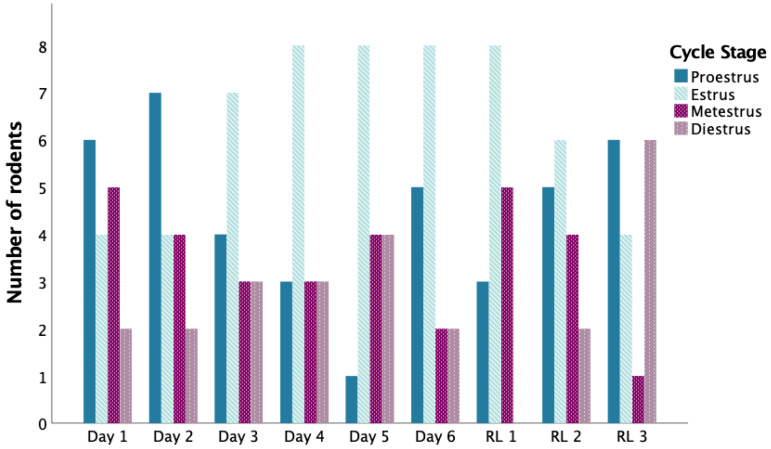
Estrous Cycle Stages. Vaginal cytological evaluation analysis was performed, and stages were visually identified for each day of the Barnes Maze Test, for adult rodents only (*n* = 17).

**Table 1 nutrients-14-03513-t001:** ANOVA Summary Table for the Acquisition Phase.

Variable	*df*	Mean Squares	*F*	*p*	Effect Size
**Latency to Escape Box**					
Day	2.935	560,387	32.893	<0.001	0.370
Trial	1	105,745	43.911	<0.001	0.440
Day × Age	2.935	65,152	3.824	0.012	0.064
Trial × Sex	1	18,534	7.696	0.008	0.121
**Distance Travelled**					
Day	3.424	70,610,325	51.542	<0.001	0.484
Trial	1	33,669,400	120.475	<0.001	0.687
Day × Sex	3.424	40,94,996	2.989	0.026	0.052
Day × Age	3.424	6,324,490	4.617	0.002	0.077
Trial × Supplementation	1	2,593,904	9.281	0.004	0.144
Day × Trial	4.351	2,302,846	4.687	<0.001	0.079
Trial × Supplementation × Sex × Age	1	3,816,087	13.655	<0.001	0.199
**Number of Working Memory Errors**					
Day	3.37	192.113	3.795	0.009	0.062
Trial	1	290.854	31.235	<0.001	0.354
Day × Age	3.37	728.147	23.095	<0.001	0.288
Day × Supplementation × Sex	3.37	92.468	2.933	0.029	0.049
Day × Sex × Age	3.37	97.631	3.097	0.023	0.052
Trial × Age	1	65.860	7.073	0.01	0.110
Day × Trial	5	32.285	2.841	0.016	0.047
Day × Trial × Age	5	44.373	3.904	0.002	0.064
Day × Trial × Supplementation × Sex	5	34.503	3.036	0.011	0.051

Note: For ease of interpretation, only significant interactions and pairwise comparisons are presented. Effect size = partial η^2^.

**Table 2 nutrients-14-03513-t002:** ANOVA Summary Table for the Reversal Learning Phase.

Variable	*df*	Mean Squares	*F*	*p*	Effect Size
Latency to Escape Box					
Day	2	107,000	24.926	<0.001	0.462
Distance Travelled					
Day	2	6,179,837	15.019	<0.001	0.357
Trial	1	1,969,308	5.734	0.024	0.175
Day × Sex	2	4,415,398	10.731	<0.001	0.284
Day × Trial × Sex	2	2,637,329	7.113	0.002	0.209
Number of Working Memory Errors					
Day	2	138.104	13.818	<0.001	0.33
Day × Trial × Sex	2	17.657	3.195	0.049	0.102

Note: For ease of interpretation, only significant interactions and pairwise comparisons are presented. Effect size = partial η^2^.

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
