# Peer review of "Short-Term Fish Oil Supplementation during Adolescence Supports Sex-Specific Impact on Adulthood Visuospatial Memory and Cognitive Flexibility"

_nutrients, 2022, doi:10.3390/nu14173513_

Round 1

Reviewer 1 Report

The study is interesting and well planned. It has significant cognitive value. The authors presented in detail the research technique they used. The results were presented with attention to detail, although in my opinion the results presented in the tables (not in the text) would be clearer. Doubts are raised by the description of the statistical analysis performed, which is inconsistent with the statistical tests to which the authors refer in the chapter on results. The authors reported that they used a five-way mixed ANOVA for the statistical analysis, while presenting the results they referred to the five-way mixed ANOVA and the four-way mixed ANOVA. In my opinion, the chapter on the statistical tests used should be clarified. The conclusions are very optimistic and general. The observed differences are minimal and differently oriented. In my opinion, the conclusions should be formulated more carefully, because the results, although promising, certainly require verification.

Overall, in my opinion, the presented study is valuable. The work can be published after making the necessary corrections.

Author Response

Reviewer 1

 The study is interesting and well planned. It has significant cognitive value. The authors presented in detail the research technique they used. Overall, in my opinion, the presented study is valuable. The work can be published after making the necessary corrections.

The results were presented with attention to detail, although in my opinion the results presented in the tables (not in the text) would be clearer.

Answer: Thank you for this suggestion. We have added a table summarizing the main effects and interactions of both phases of the Barnes maze test to lighten the text (please see Table 1 and Table 2). We opted to keep the pairwise comparisons written out, for clarity and to avoid making the tables too large.

Doubts are raised by the description of the statistical analysis performed, which is inconsistent with the statistical tests to which the authors refer in the chapter on results. The authors reported that they used a five-way mixed ANOVA for the statistical analysis, while presenting the results they referred to the five-way mixed ANOVA and the four-way mixed ANOVA. In my opinion, the chapter on the statistical tests used should be clarified.

Answer: Thank you for this observation. We have modified the statistical analyses section to clarify this point: “For the acquisition phase, five-way mixed analyses of variance (ANOVA) were performed using IBM SPSS Statistics 28.0 software with between-group factors Supplementation (FO vs CSO), Age (juvenile vs adult), and Sex (female vs male) as well as within-group factors Day (1-6) and Trial (1 vs 2). Four-way mixed ANOVA was performed for reversal learning trials, with Supplementation (FO vs CSO) and Sex (female vs male) as between-group factors and Day (1 to 3) and Trial (1 vs 2) as within-group factors.”

The conclusions are very optimistic and general. The observed differences are minimal and differently oriented. In my opinion, the conclusions should be formulated more carefully, because the results, although promising, certainly require verification.

Answer: Thank you. We have adjusted the conclusions to reflect the scope of our findings more accurately.

Miscellaneous changes: Section 3.2 (WME) was changed with section 3.3 (Distance travelled) for consistency with the reversal learning trials sections.

Reviewer 2 Report

Overall great job. The methods and writing are appropriate. You do not need to describe the Shapiro-Wilks test and Levene's test in the methods section unless you found the data was NOT normally distributed. You can remove that as extraneous.

Also, I did not see any information on potential strengths & limitations described in the discussion section.

Author Response

Reviewer 2

Overall great job. The methods and writing are appropriate. You do not need to describe the Shapiro-Wilks test and Levene's test in the methods section unless you found the data was NOT normally distributed. You can remove that as extraneous.

Answer: We have removed the unnecessary information.

Also, I did not see any information on potential strengths & limitations described in the discussion section.

Answer: Thank you. We have added a limitations section at the end of the discussion (please see section 4.5).

Miscellaneous changes: Section 3.2 (WME) was changed with section 3.3 (Distance travelled) for consistency with the reversal learning trials sections.